# Methodology for Earthquake Rupture Rate estimates of fault networks: example for the Western Corinth Rift, Greece

Thomas Chartier[1,2], Oona Scotti[2], Hélène Lyon-Caen[1] , Aurélien Boiselet[1,2*]

[1] Laboratoire de géologie, Ecole Normale Supérieure, CNRS UMR 8538, PSL Research University, Paris, 75005, France
[2] Bureau d'Evaluation des Risques Sismiques pour la Sûreté des Installations, Institut de Radioprotection et de Sûreté Nucléaire, Fontenay-aux-Roses, France
* now at Axa Global P&C, Paris, 75008, France.
*Correspondence to*: Thomas Chartier (chartier@geologie.ens.fr)

**Abstract.** Modelling the seismic potential of active faults is a fundamental step of probabilistic seismic hazard assessment (PSHA). An accurate estimation of the rate of earthquakes on the faults is necessary in order to obtain the probability of exceedance of a given ground motion. Most PSHA studies consider faults as independent structures and neglect the possibility of multiple faults or fault segments rupturing simultaneously (Fault to Fault -FtF- ruptures). The latest Californian model (UCERF-3) takes into account this possibility by considering a system level approach rather than an individual fault level approach using the geological, seismological and geodetical information to invert the earthquake rates. In many places of the world seismological and geodetical information along fault networks are often not well constrained. There is therefore a need to propose a methodology relying on geological information alone to compute earthquake rates of the faults in the network. In the proposed methodology, a simple distance criteria is used to define FtF ruptures and consider single faults or FtF ruptures as an aleatory uncertainty, similarly to UCERF-3. Rates of earthquakes on faults are then computed following two constraints: the magnitude frequency distribution (MFD) of earthquakes in the fault system as a whole must follow an a-priori chosen shape and the rate of earthquakes on each fault is determined by the specific slip-rate of each segment depending on the possible FtF ruptures. The modelled earthquake rates are then confronted to the available independent data (geodetical, seismological and paleoseismological data) in order to weight different hypothesis explored in a logic tree.

The methodology is tested on the Western Corinth Rift, Greece (WCR) where recent advancements have been made in the understanding of the geological slip rates of the complex network of normal faults which are accommodating the ~15 mm/yr North-South extension. Modelling results show that geological, seismological and paleoseismological rates of earthquakes cannot be reconciled with only single fault rupture scenarios and require hypothesising a large spectrum of possible FtF rupture sets. Furthermore, in order to fit the imposed regional Gutenberg-Richter MFD target, some of the slip along certain faults needs to be accommodated either with interseismic creep or as post-seismic processes. Furthermore, computed individual fault's MFDs differ depending on the position of each fault in the system and the possible FtF ruptures associated with the fault. Finally, a comparison of modelled earthquake rupture rates with those deduced from the regional and local earthquake catalogue statistics and local paleoseismological data indicates a better fit with the FtF rupture set constructed with a distance criteria based on a 5 km rather than 3 km, suggesting, a high connectivity of faults in the WCR fault system.

## 1 Introduction

The goal of probabilistic seismic hazard analysis (PSHA) is to estimate the probability of exceeding various ground-motion levels at a site (or a map of sites) given the rates of all possible earthquakes. The first step of PSHA following a Cornell-McGuire (Cornell 1968, McGuire 1976) approach is the characterization of the seismic sources. For regions where active faults have been identified and their slip-rates are known, several methods have been proposed in order to calculate the rate of earthquakes occurring on these faults. The most commonly used methods consider faults as independent structures on

which the strong earthquakes are located (e.g., SHARE Project in Europe Woessner et al, 2015; Yazdani et al, 2016 in Iran; TEM Model in Taiwan Wang et al, 2016 ). In these PSHA studies, a background seismicity will generate earthquakes up to a threshold magnitude of 6.0 or 6.5, beyond which earthquakes are generated on the faults. The rate of earthquakes for these larger magnitudes is based on geological and paleoseismological records, and the maximum magnitudes depend on the physical dimensions of the fault under consideration. In the resulting model, the rate of lower magnitudes is controlled by seismological information and the rate of stronger magnitudes by geological information. In cases where large historical earthquakes are associated to multiple fault segments, the individual fault segments described by the geologists in the field are regrouped in a larger fault source and a mean slip rate is attributed to the fault source. A specific magnitude-frequency distribution (MFD), often Gutenberg-Richter (GR) (Gutenberg & Richter, 1944) or Characteristic Earthquake (Wesnousky, 1986), describing the mean slip-rate based earthquake rate on the fault is attributed to each fault source. This process requires simplifying fault complexity in terms of geometry and slip-rate and doesn't allow complex ruptures that propagate from one fault source to an adjacent one.

In the past decades, the quality of the observation has improved and our understanding of earthquakes has grown (Proceedings of the 2017 Fault2SHA Meeting). We observe more and more complex earthquake ruptures propagating on several neighboring faults. There is thus a need for hazard models to accurately represent the faults and ruptures complexity observed in the field by geologists and to correctly distinguish aleatory from epistemic uncertainties.

Toro et al. (1997) define epistemic uncertainty as "uncertainty that is due to incomplete knowledge and data about the physics of the earthquake process. In principle, epistemic uncertainty can be reduced by the collection of additional information." Aleatory uncertainty on the other hand, is an "uncertainty that is inherent to the unpredictable nature of future events" (Toro et al 1997); in this respect fault-to-fault (FtF) rupture should be treated as an aleatory uncertainty since it is linked to the randomness of the seismic phenomenon.

In order to allow FtF ruptures, the Working Group on California Earthquake Probabilities (WGCEP-2003) for San Francisco Bay Region developed a methodology that explores possible FtF ruptures in a logic tree. Each branch of the logic tree represents a seismic hazard model and the rate of the corresponding FtF rupture scenario is obtained by weighting the branches. Gulerce & Ocak 2013 used this approach and set the weight of each branch (or rupture scenario) such that the mean seismicity rate modeled by the logic tree fits the recorded seismicity rate around the fault of interest. This method treats the uncertainty of FtF ruptures as an epistemic uncertainty in the PSHA calculation.

The latest Californian model UCERF-3 was developed using a novel methodology that treats all possible combinations of FtF rupture scenarios within the same branch of the logic tree as an aleatory uncertainty (Field et al, 2014). In their terminology, faults are divided in smaller sections and all possible section-to-section ruptures are investigated. The possibility of ruptures happening is controlled by a set of geometric and physical rules and the rate of earthquakes is computed using a "grand inversion" of the seismological, geological, paleosismological and geodetic data available in California. The regional GR MFD of earthquakes of California and the geodetic deformation are used as a target for the total earthquake rupture forecast in each deformation model. This grand inversion relies also on estimates of the creep rate on faults deduced from local deformation data when available.

For many fault networks, only sparse seismological and geodetic data are available and the geological record is often the most detailed source of information concerning the faults' activity. In such cases, it's necessary to develop a methodology that allows building seismic hazard models relying only on geological data and yet allowing FtF ruptures as an aleatory uncertainty. The sparse geodetical, seismological and paleoseismological data can then be used as a means of comparison to help weighting the different input hypothesis.

In this study we propose such a methodology based on slip-rate budget of each fault, FtF ruptures hypothesis and assumptions on the shape of the MFD defined for the fault system as a whole. The methodology is developed so as to be flexible and applicable to regions where data on faults, geodesy and seismicity may be sparse. The rate of earthquakes on

faults computed using geological information (slip-rates) is then compared to other sources of information such as the regional and local earthquake catalogues and the paleoseismic data in order to weigh the different epistemic uncertainty explored in the logic tree. Moreover, it is also known that faults accommodate important amounts of slip in either post-seismic slip or in creep events (e.g. L'Aquilla 2009, Cheloni et al., 2014, Napa earthquake 2014, Lienkaemper et al., 2016).

These phenomena, called Non-Main-Shock slip (NMS) later on, are integrated in the slip-rates deduced from geological information and should not be converted into earthquake rates when computing seismic hazard. The methodology presented in this study allows part of the geological slip-rate to be considered as NSM slip-rate.

We use this methodology to generate fault-based hazard models for the Western Corinth Rift (WCR), Greece, which has been studied for the past decade by the Corinth Rift Laboratory Working Group (CRL-WG) (Lyon-Caen et al, 2004; Bernard

et al, 2006; Lambotte et al, 2014). A large number of active faults have been identified in this area and a consensus about their possible geometries and activity rates has been reached within the CRL-WG (Boiselet, 2014). We used this geologic information to test our modelling approach and explore different epistemic uncertainties in a logic tree. Finally, we confront the modelled earthquake rates of each fault with seismological and paleoseismological data in order to weigh the hypothesis in the logic tree.

## 15 2 Novel methodology for taking faults into account in PSHA

In most regions of the world the amount of data available to model faults in a PSHA study is often sparse and uncertain. However, the need to consider such data in PSHA is increasing and the methods to properly incorporate the available geological information in the hazard models are still missing. In this context, we propose to build a methodology that allows considering all the available information on faults, allows setting rules to define FtF ruptures and considers single faults or

20 FtF ruptures as an aleatory uncertainty.

Our iterative method allows converting the slip-rate budget of each individual fault rates of earthquakes by imposing that the resulting regional MFD of earthquakes in the whole fault system follows an imposed shape. The MFD shape of each individual fault will thus be a result of the iterative process and not an imposed parameter.

The proposed method is presented here in a nutshell and illustrated in Figure 1.

(1) The necessary input data includes:
  - a definition of the 3D geometry of the fault system.
  - an estimate of the geological slip rates of each individual fault that determines the slip-rate budget of the fault.

(2) Setting up the methodology requires:

- choosing suitable scaling laws to estimate the maximum magnitude each fault can host.
  - assuming minimum magnitude of earthquakes possible on the faults (5.0 in this study).
  - hypothesizing FtF rupture scenarios based on some rules. In this study only a simple distance rule is used to define FtF ruptures. In future developments, more physics based approaches could be explored. In the example presented in Figure 1a, the three faults (*fault 1, fault 2* and *fault 3*) are considered to be sufficiently close to each other thus

they can either rupture individually (F1, F2, F3) or in FtF rupture (F1+F2, F2+F3 or F1+F2+F3).
  - imposing a shape for the target MFD for the whole fault system. In this study a GR MFD distribution is assumed.

(3) Two pre-computational steps are performed to:
  - calculate all possible magnitude bins each fault and FtF rupture scenario can accommodate according to each scaling law considered (Figure 1b).

- calculate the number of incremental quantities of slip-rate (*dsr*) contained in each fault budget. In the example, the *fault 1, fault 2* and *fault 3* have a slip-rate budget of 5 mm/yr, 3.2 mm/yr and 4 mm/yr, respectively (Figure 1d).

Therefore considering a dsr size of 0.01 mm/yr, the faults budgets will be consumed after 500, 320 and 400 dsr respectively. The slip-rate budget of each fault may be spent along individual faults and/or FtF rupture scenarios.

- convert the target MFD, expressed in terms of rate of earthquakes, into moment rates (Figure 1 b). This target MFD will be used to pick the magnitude bin on which an increment *dsr* will be spent on. Notice that the formulation in terms of moment rate implies that greater magnitudes are more likely to be picked.

(4) Iterative process :

❖ First, the bin of magnitude (of width 0.1) where a *dsr* will be spent is picked according to the target MFD for the whole fault system in terms of moment rate.

❖ Then, in this bin of magnitude $M_i$, a seismotectonic source *Si* (an individual fault or an FtF scenario) that can host this magnitude is picked randomly. The increment of moment rate $d\dot{M}_0$ for this source is calculated following equation 1 and the rate of earthquakes increment $dr_e$ is calculated using equation 2.

$$dM_0 = \mu . A . dsr \tag{1}$$

$$dr_e(M_i) = \frac{d\dot{M}_0}{M_0(M_i)} \tag{2}$$

where $d\dot{M}_0$ is the increment of moment rate for the source $S_i$, μ the shear modulus of the fault, A the area of the source, *dsr* the increment of slip-rate spent, $dr_e(M_i)$ the increment of the rate of magnitude $M_i$ and $M_0(M_i)$ the seismic moment of a moment magnitude *M* defined by Hanks and Kanamori (1979):

$$M = \frac{2}{3}\log(M_0) - 10.7 \tag{3}$$

❖ At each iteration, the slip-rate budget of the faults participating to the scenario accommodating the earthquakes of the three highest magnitude bins (0.3 being the range of uncertainties in the scaling laws used to assess the maximum magnitude) is checked:

➢ If there is still slip-rate budget to be spent, the $dr_e$ calculated is added to the rate of earthquakes of magnitude $M_i$ for the source $S_i$.

➢ If one of the faults of the FtF rupture generating the largest earthquake has exhausted its slip-rate budget, the final rates of the highest magnitude earthquakes is reached. Then knowing the shape of the imposed target MFD, the target rate at the fault system level for all magnitudes bins is known (Figure 3c). At this stage, an additional check is made :

- if adding the $dr_e$ calculated for magnitude $M_i$ on the source $S_i$ leads to exceed the target MFD for this magnitude, then this $dr_e$ is not added to the source $S_i$ and the increment *dsr* of this computation step is considered as Non Main-Shock (NMS) slip.

- if adding the $dr_e$ to the source $S_i$ does not lead to exceed the target MFD, this $dr_e$ is added to the source $S_i$.

❖ The increment of slip-rate *dsr* is then removed from the slip-rate budget of the fault or the faults involved in source $S_i$.

❖ If the fault's slip-rate budget of each fault is exhausted, the fault and the corresponding FtF rupture scenarios the fault is involved in are removed and cannot be picked anymore in subsequent iterations of the computation.

❖ These steps are repeated until all the slip-rate budgets of all the faults in the system is spent either on single fault ruptures, FtF ruptures or NMS slip .

The output of this process is an earthquake rupture rate for different magnitudes for each fault and FtF rupture scenario in the model, considered as aleatory uncertainty. We also record how the slip-rate budget of each fault is partitioned between the different FtF ruptures and how much NMS-slip was needed on each fault in order to fit the target MFD shape (here GR MFD) with a given set of FtF rupture scenarios (Figure 1 d). In the example (Figure 1d), *fault 1* spends 43% of its budget on

single fault ruptures (blue color), 7% on F1+F2 ruptures (dark green), 23% on F1+F2+F3 ruptures (dark grey) and 27% on NMS slip (light grey). On the other hand, the slip-rate budgets of the slower moving fault (i.e. *fault 2*) is converted 100% into earthquake rates (0% NMS) and limits the rate of the largest magnitude earthquakes (F1 + F2 + F3) (see Supplementary material).

A simplified example of application of this methodology based on only two faults is given as an annex to this paper. This example illustrates step-by-step the way in which the proposed methodology allows to transform slip-rate budgets of faults into earthquake rates.

Post-processing then includes:

   ❖ Exploring the epistemic uncertainties:

Many assumptions have to be made when setting up the methodology (scaling law, FtF rupture set, faults parameters …) and the different possible hypothesis should be explored in a logic tree.

   ❖ Reality checks :

       The last step of the methodology involves comparing the modeled earthquake rates with independent data such as the seismicity rates deduced from the catalogue and from paleoearthquake rates deduced from trench studies. Each

branch of the logic tree is then weighted according to its performance with this independent data.

 In this study, we applied the proposed methodology to a well-documented rifting zone in the North of the Peloponnese, Greece.

**3 Application to the western Corinth rift fault system**

The East West striking Corinth Rift is the most seismically active structure in Europe with several earthquakes larger than

5.5 recorded in the historical times as well as in the instrumental period (e.g. Jackson et al, 1982; Papazachos and Papazachou, 2003; Makropoulos et al, 2012). The Corinth Rift Laboratory (CRL) was set up in 2001 in the western and most seismically active part of the rift (Lyon-Caen et al, 2004) with the goals of understanding the rifting process and providing key elements for the seismic hazard assessment of the region.

The geodetic deformation measured by Global Positioning System (GPS) shows a highly localized opening of the Corinth

Rift at a rate of 10 mm/yr in the east and 15 mm/yr in the west (Avallone et al. 2004) over a distance of around 20 km inducing a high strain-rate. This deformation is accommodated by a complex network of both north and south-dipping normal faults. Geological studies of these faults have shown that the north dipping faults located on the southern coast have a higher slip-rate than the south-dipping northern faults, giving the rift its asymmetrical structure. In the south, the Peloponnese is uplifted by the activity of these faults (Armijo et al., 1996, Ford et al., 2013) and in the north the coast line is

subsiding.

The Western Corinth Rift (WCR)  faults slip-rates were inferred from the displacement of geologic markers in the field or from seismic profiles on each individual fault with the exception of the two blind faults identified by their recent seismic activity (1995 fault, Bernard et al., 1997  and Pyrgos fault, Sokos et al., 2012) and for which the microseismicity recorded close to the fault was transformed into slip-rate on the fault plane. These latter slip-rates are therefore subject to a very large

uncertainty. The estimated geological extension rate expressed by the sum of the horizontal projection of the geological slip-rates of the faults is in the range of 3 to 6 mm per year, three times less than the geodetic extension rate. Given this disagreement, the WCR is a good candidate to test if the earthquake rates calculated using our methodology that relies only on geological information can account for the occurrence of large earthquakes that have been observed in the region (Albini et al 2017).

The WCR fault system has been described by Boiselet 2014, defining a model for the fault system, including geometries and slip-rates for each fault (Figure 2, Table 1) and a set of possible FtF ruptures (hereafter model B14). The B14 model

proposes a set of FtF rupture scenarios (Table 2) assuming that two neighboring faults can make up a FtF scenario only if they are less than 3 km apart. In this paper, we also explore a logic tree branch for an alternative rupture set (see Figure 3) with higher fault connectivity (B14_hc) where faults can break together if their fault traces are separated by 5 km or less, therefore allowing a wider spectrum of possible FtF rupture scenarios (additional scenarios in bold in Table 2). As a comparison with classical fault PSHA studies, we explore a branch with only simple fault rupture called B14_s. In this branch no FtF rupture is allowed.

The target MFD shape is chosen to be a GR with a b-value of $1.15 \pm 0.05$ which is a typical value for extensional systems (Schorlemmer et al, 2005).

In this study we explore other epistemic uncertainties having potentially an impact on the modelled earthquake rates (Figure 3) in addition to the different FtF rupture sets previously described, two scaling laws (Wells and Coppersmith 1994 WC94 and Leonard 2010 Le10) have been used to calculate the maximum magnitude based on rupture area for normal faults, and two values of the shear modulus µ: 30 GPa, commonly used value in hazard studies, and 20 GPa to represents the low shear waves velocity in the WCR region recently estimated based on ambient noise tomography (Giannopoulos et al, 2017). For each branch, 20 random samples are drawn from triangular distributions in order to explore the epistemic uncertainty affecting the b value of the target MFD ($1.15 \pm 0.05$), the slip-rate of the faults and the uncertainty within the scaling law.

**4 Modeled earthquake rupture rates and comparison with independent data**

Using our method, the modeled the rate of earthquakes for the WCR is then compared to the rate of earthquakes observed in the catalogue. The seismicity catalogue considered in this study (Figure 2) is the SHEEC catalogue (Giardini et al., 2013; Stucchi et al., 2012; Grünthal et al., 2013) developed in the framework of the SHARE project updated for 6 historical earthquakes (Albini et al., 2017) and 3 instrumental earthquakes (based on Baker et al 1997 study and personal communication from the 3-HAZ Corinth project). The updates and their implication on the catalogue are summarized in Table 3. We propagate the earthquake magnitude uncertainties in the estimate of seismic moment rate and earthquake rate calculations by randomly sampling the magnitude of each earthquake within their uncertainties (Stucchi et al, 2012, Albini et al, 2017) and by using two hypotheses of completeness. In Table 4 the times of completeness for Greece calculated by the SHARE project (Stucchi et al., 2012) and the times calculated by Boiselet 2014 using the Stepp 1972 approach at the scale of the Corinth Rift region are reported.

A first reality check is to compare the modeled and the catalogue seismological moment rates. The seismological moment rate is calculated directly using the rates of earthquake of each magnitude in the catalogue based on the moment magnitude relation (equation 3) The seismic moment rate in models B14 and B14_hc are in good agreement with the seismic moment rate deduced from the catalogue whereas the B14_s predicts a higher seismic moment rate (Figure 4a). This comparisons brings a better confidence in the models where FtF ruptures are possible than in the B14_s model. In the single-rupture model (B14_s), 90% to nearly 100 % of the geological slip-rate is converted into seismic moment rate with only less than 10% interpreted as NMS slip-rate. On the other hand, when FtF ruptures are possible (B14 and B14_hc), 25% of the geological slip-rate budget of the faults is interpreted as NMS slip (Figure 4b).

A second reality check consist in comparing modeled and catalogue MFDs (Figure 4c). The B14_s model doesn't manage to reproduce the rate of earthquakes deduced from the catalogue, as it predicts a higher rate of magnitude 5 earthquakes and no earthquakes of magnitude 6.3 and above. On the other hand, we observe a good agreement of the MFDs of models B14 and B14_hc with the catalogue. B14 reproduces well the cumulative earthquake rate for magnitude 5.6 to 6.1 whereas model B14_hc reproduces better the cumulative rate of earthquakes of magnitude 5.0 to 5.5.

Slip-rate budget repartition

The way the slip-rate budget is spent between FtF rupture and single fault rupture and the NMS slip ratio of the fault depends on the slip-rate of the fault and the FtF ruptures the fault is involved in. Slow slipping faults that are involved in large FtF rupture scenario (Neos-Erineos or West Helike) have the majority of their slip-rate budget consumed by these large FtF ruptures (Figure 5). On the contrary, the fast slipping faults that are involved in few FtF ruptures scenarios (1995, Pyrgos, North-Eratini) spend their budget on predominantly single fault ruptures producing a high number of small to medium magnitude earthquakes which lead to easily exceed the GR regional target and thus imply a higher proportion of NMS slip-rate on these faults.

Models B14 and B14_hc have a similar mean 25% ratio of NMS slip ratio (Figure 4) but this ratio is not distributed between the different faults in the same way in each model. An important NMS proportion on the blind faults (Pyrgos and 1995-fault) and the off-shore North-Eratini fault is found for both models. There are three main factors that can induce this result: either the FtF sets are not realistic, the slip-rates explored on those faults are not realistic and don't include enough complex ruptures with these faults, or there is a mechanism of NMS slip such as creep or slow slip events happening on theses faults.

Earthquake rupture rate on the Aigion Fault

We choose now to focus our interest on the Aigion fault. Since this fault is one of the most active faults of the WCR and crosses the city of Aigion, it represents a major source of seismic hazard and risk for the region.

The earthquake rate modelled on the Aigion fault depends of the FtF rupture set allowed in the model (Figure 6). The resulting MFD of the Aigion fault has the shape of a GR for model B14 and B14_s, with a steeper slope for the B14_s model. In the B14_hc, the MFD computed for the Aigion fault is more similar to a Characteristic Earthquake of magnitude close to 6.0, similar to the maximum magnitude of earthquakes rupturing only the Aigion fault. It is worth noting that the larger magnitude earthquakes in Figure 6b and c involve not only the Aigion fault but also the neighboring faults participating in the FtF ruptures (Figure 5, Table 2).

Using the paleoseismological data presented by Pantosti et al 2004, it is possible to propose rates of large magnitude earthquakes on the Aigion fault (figure 6). This paleorate is subject to large uncertainties but can be used to validate or invalidate the different FtF rupture set hypothesis. In the B14_s model where faults only break on their own, the Aigion fault is not able to accommodate the paleo-earthquake magnitudes. In the B14 model, where fault rupture is only allowed between faults separated by 3 km or less, the modelled earthquake rates are lower than the rates inferred from the paleoseismological study. In the B14_hc model, where FtF ruptures are allowed for faults separated by 5km or less, the modelled earthquake rates agree well with the paleorate, within the margin of uncertainty.

According to the recent reappraisal of the historical seismicity (Albini et al., 2017), the Aigion fault is most likely the source of the 1817 M 6.5 [6.0-6.5] and the 1888 M 6.2 [5.7 – 6.2] earthquakes. This leads to estimates of annual rates of M>6 earthquakes on the Aigion fault of 0.005 to 0.007 (Figure 6) depending on the completeness period used (Table 4). The model B14_s doesn't manage to reproduce the great magnitudes earthquakes observed in the catalogue. The annual rates for earthquakes M>6 of 0.0034 and 0.0051 predicted by models B14 and B14_hc respectively are statistically compatible with the rate inferred from the catalogue.

Weighting the logic tree branches

The comparison with independent local data allows suggesting weights for the different FtF rupture set hypothesis (Figure 3) for hazard calculation.

The B14_s branch, where faults can only rupture independently does not fit neither the annual moment rate nor the earthquakes rate of the catalogue of the region, nor the paleoearthquake magnitude on the Aigion fault (Figure 4 and 6). We conclude that this branch should not be used for a hazard calculation in the Western Corinth Rift.

Between the two branches where FtF ruptures are possible, B14_hc manages to match the earthquake rates of the catalogue for a range of magnitudes where statistics are stronger (14 earthquakes of magnitude 5.0 and above) compared to the B14 model (matching only 4 earthquakes of magnitude 6.0 and above in the catalogue) (Figure 4). B14_hc branch matches the Aigion fault earthquake rates inferred from the paleoseismology and the historical catalogue better than the B14 model (Figure 6). The agreement with the earthquake rate in the regional catalogue and the better reproduction of the Aigion fault data of the B14_hc model leads us to propose a stronger weight for this model compared to the B14 model for the estimate of hazard for Aigion city.

## 5 Conclusion and Perspectives

The methodology presented in this study uses a system level approach rather than an individual fault level approach to estimate the rate of earthquakes on faults based on the geological data collected for each fault and allowing FtF rupture in the hazard model as an aleatory uncertainty. The application of the methodology to the WCR fault network shows that in order to match a GR MFD for the whole fault system, part of the fault slip-rates have to be spent as Non-Main-Shock slip. The way the fault slip-rate is partitioned among single or FtF ruptures and the resulting shape of the individual fault MFD depends on the location of the fault in the network and the fault's characteristics. The earthquake rates modelled using the geological data on the faults are compared with the local earthquake catalogue and paleoseismic data in order to weight the different epistemic hypothesis. In the case of the WCR, and for future seismic hazard assessment for the city of Aigion, these reality checks suggest attributing a stronger weight to the branch allowing FtF ruptures between faults with the 5 km distance criteria (B14_hc), a lower weight to that based on the 3 km criteria (B14) and a null weight to the model where only single fault ruptures are allowed (B14_s).

The fault network used for the application concerns only the western part of the Corinth Rift fault network. Integrating the rest of the network in the model could modify the final outcome and should be explored in future developments.

More reality checks will be implemented in the future in order to weigh the different uncertainties of the logic tree based on the results of the ongoing microseismicity studies in the WCR (i.e. use the possible presence of repeater earthquakes on the Aigion fault to validate NMS slip ratio, Duverger et al., 2015).

The methodology presented in this article can be applied to other fault systems, in different tectonic environments. In order to implement this approach, the geometries and slip-rates of the faults have to be known within uncertainties, FtF rupture scenarios sets have to be defined and the shape of the regional MFD needs to be assumed or inferred from the regional catalog. If for the WCR the GR distribution seems suitable, it has been shown that a Youngs and Coppersmith distribution (Youngs and Coppersmith, 1985) can be more appropriate for other fault systems (e.g. by Hecker et al., 2013). Given the flexibility of our methodology any other target MFD can be easily implemented in the methodology.

The earthquake rupture rate calculated using this methodology is very sensitive to the choice of possible FtF rupture scenarios. The comparison with the earthquake catalogue and local data, such as the paleoseismological data, can provide guidance to the strength of each hypothesis. Nevertheless, the choice based on distance between faults should be supported by more physical approaches in the future such as Coulomb stress modeling (Toda et al., 2005) and/or dynamic modelling of ruptures (Durand et al., 2017).

The methodology at this stage doesn't consider the background seismicity. The example of the dense WCR fault system allowed setting aside this issue in order to test our methodology and focus on the FtF ruptures. Future developments of the methodology need to allow part of the modelled seismicity rate to be in the background. If performing hazard calculation for a region wider than the fault system itself, it is necessary to combine the models built with this methodology with classical area sources.

## Acknowledgement

Many thanks to Kuo-Fong Ma and an anonymous referee as well as reviewer Laura Peruzza for their valuable questions and comments that helped improving the quality of this article to a great extent. This work was jointly funded by IRSN and ENS (LS 20201/CNRS 138701) and Axa Research Fund (Axa – JRI – 2016). The code used in this study is still under development but can be shared on demand.

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

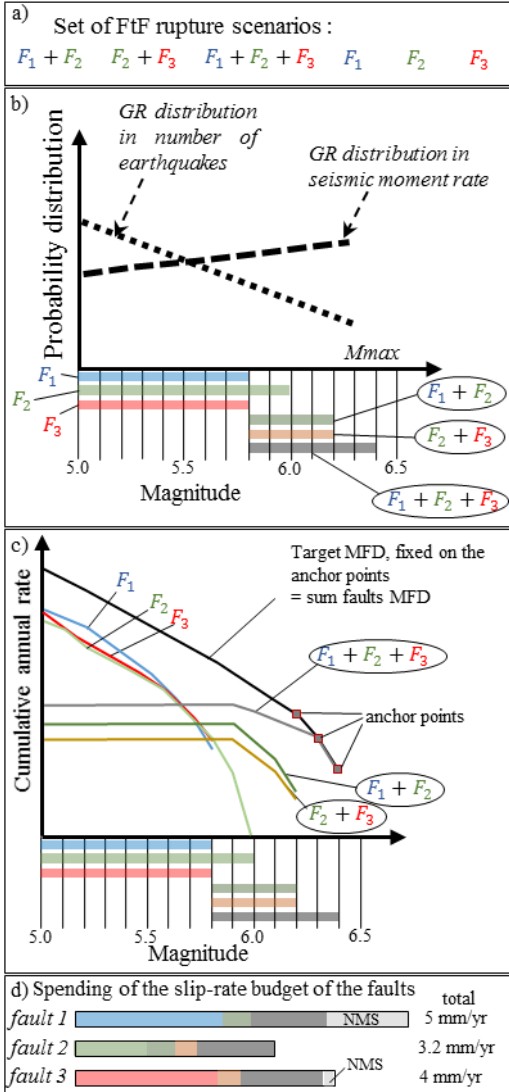

**Figure 1 : Illustration of the methodology. a) Set of FtF rupture scenarios. b) Picking of the magnitude bins and of the sources. c) Building the target MFD : the black curve is the target MFD anchored at the mean of the three highest magnitude bins (magnitude bin of 0.1). The sum of the resulting MFDs of the six sources has to be equal to the target MFD. d) visualization of the partitioning by the iterative methodology of each fault's slip-rate budget (colors correspond to the individual rupture or the FtF rupture, NMS : Non Main Shock slip).**

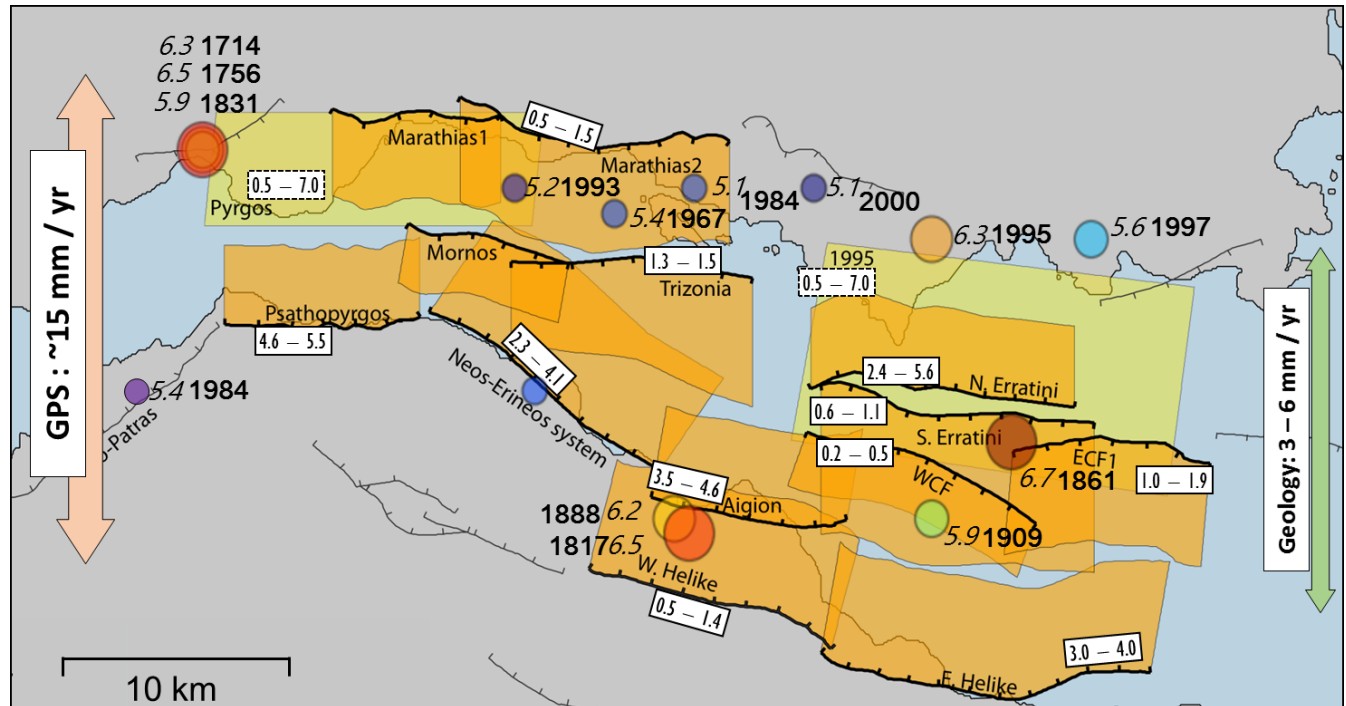

**Figure 2 : Map of the active faults of the western part of the Corinth Rift (modified from Boiselet 2014). The orange polygons are the projections to the surface of the active faults. The yellow polygons are the projection to the surface of the blind faults (Pyrgos fault and 1995 fault). Earthquakes of the catalogue during the complete period represented by the circles with color and size depending on the magnitude. Year and preferred magnitude of earthquake indicated. The minimum and maximum values (mm/yr) of the slip-rates of the faults are indicated in the white boxes. The green arrow shows an approximation of the rift extension calculated by projecting horizontally the faults slip-rate and the pink arrow shows the extensional rate of the rift measured by GPS.**

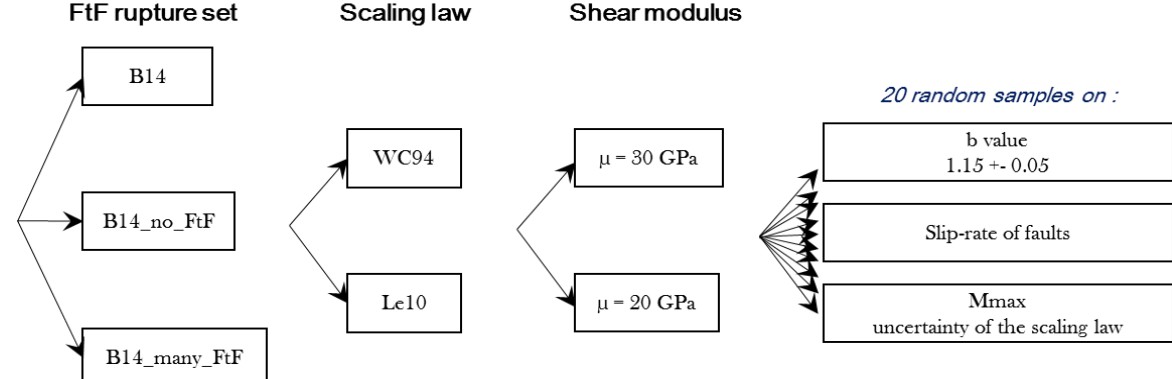

**Figure 3 : Logic tree explored for this study**

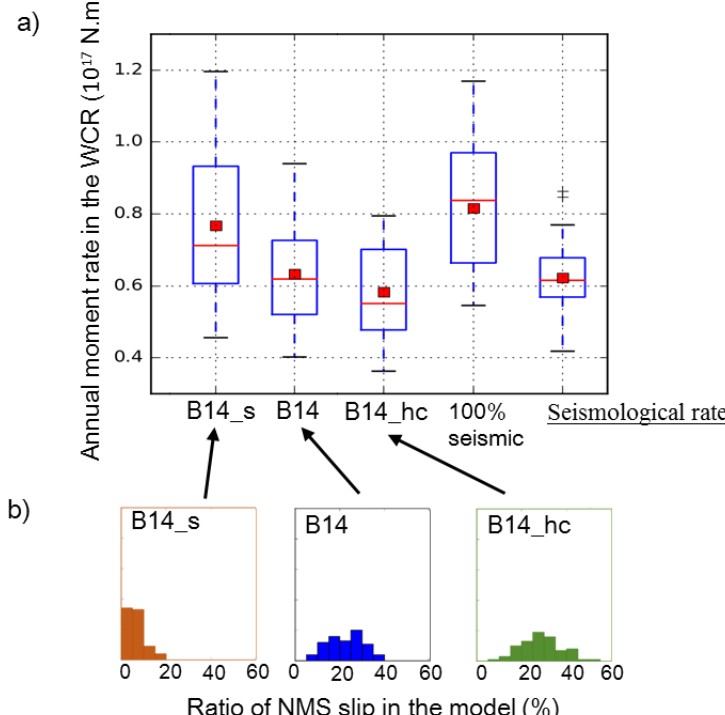

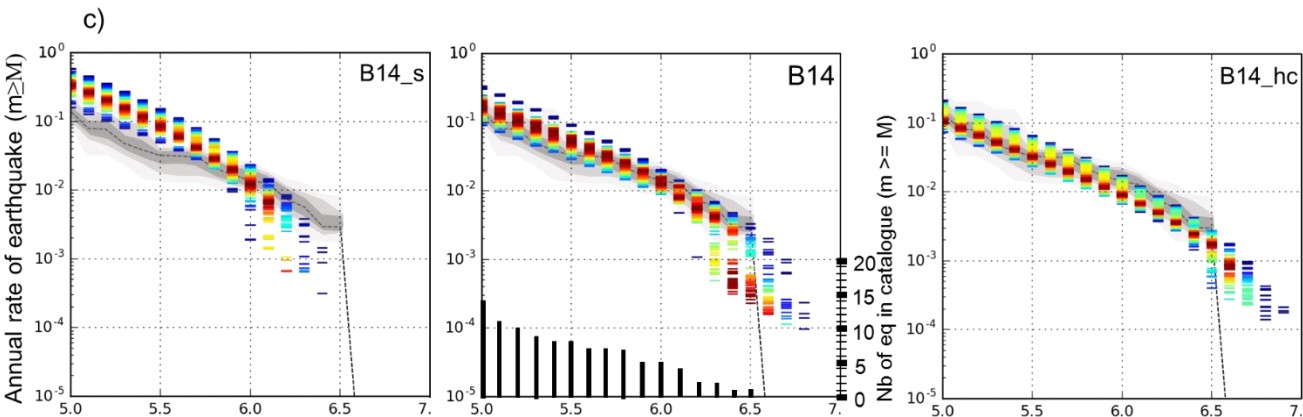

**Figure 4 : Modelled seismicity for the WCR fault network and comparison to the seismicity rate based on the earthquake catalogue of the complete period. a): comparison between the modelled moment rates for each FtF scenario set and the seismological rate calculated from the earthquake catalogue. Each box represents the standard deviation around a mean and median value represented by a red square and a red line respectively. From left to right: the three first boxes are for each hypothesis of scenario set in the logic tree, the fourth box shows the moment rate assuming 100% of the slip-rate of faults is converted into seismic moment, the fifth box shows the moment rate calculated from the earthquake catalogue. b): distribution of the ratio of NMS slip resulting from the three deformation models. c): Comparison between the modelled GR MFD deduced from geological data for the whole fault system and that deduced from the WCR catalogue. The models are represented as a colored density function with the red colors for the rates predicted by the higher number of models. The cumulative rates calculated from the catalogue are shown as a grey density function. The cumulative number of earthquakes in the catalogue is indicated by black bars in the central figure.**

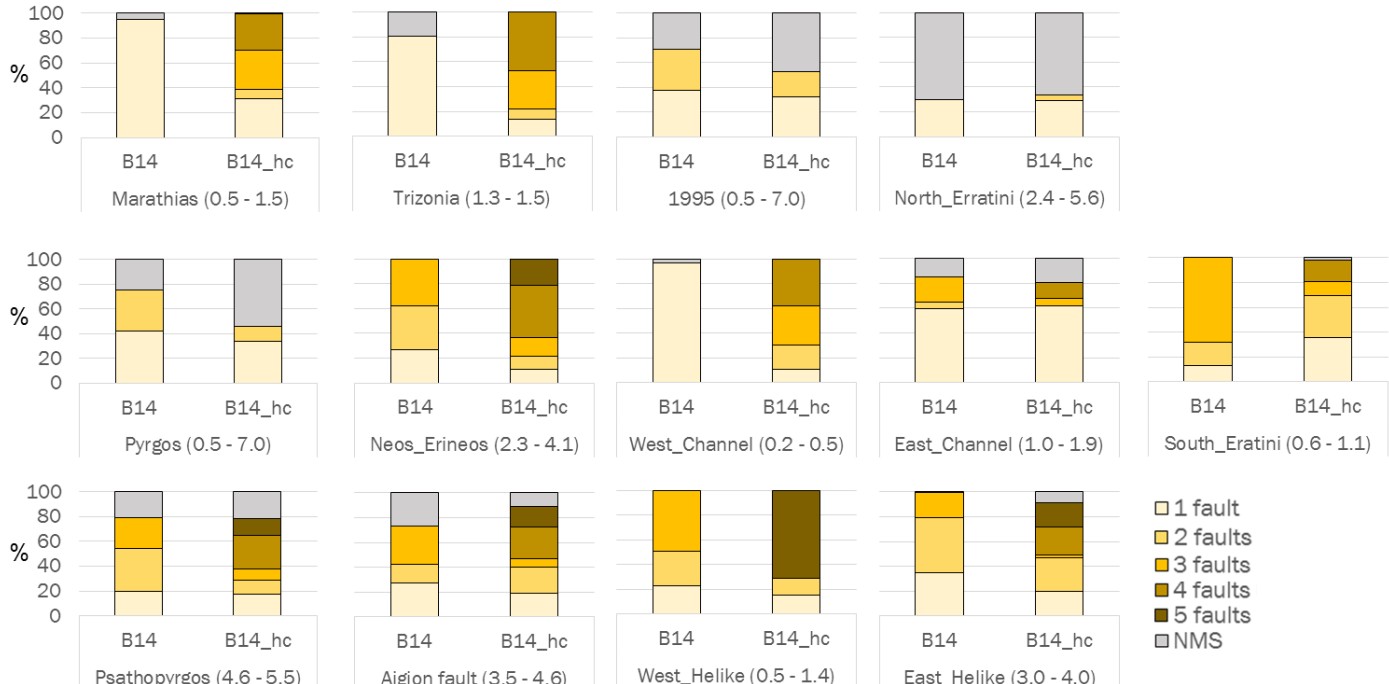

**Figure 5 : Visualization of the way the slip-rate budget of each fault is spent. The color depends on the number of faults involved in the FtF rupture. Minimum and maximum values of the slip-rate on each fault is shown in brackets in mm/yr.**

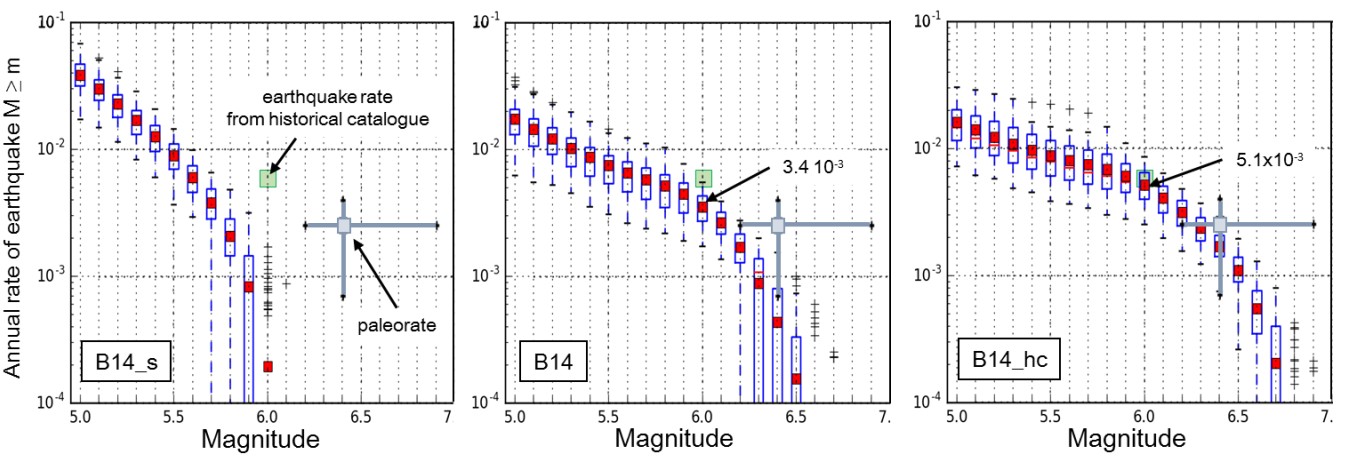

**Figure 6 : Rate of earthquakes occurring on the Aigion fault for each FtF rupture set. Variability resulting from the exploration of the logic tree is illustrated by the blue boxes. The annual rates of M≥6 earthquakes on Aigion fault is indicated for the B14 and B14_hc models. The grey square represents the paleorate interpreted from Pantosti et al. (2004) and its uncertainties. The green box represents the rate of earthquakes greater than magnitude 6 on the Aigion fault inferred from the historical catalogue.**

| fault name | ID | Length | dip | seismogenic depth (km) | | slip-rate (mm/yr) | | | Mmax | | time frame of the data |
|---|---|---|---|---|---|---|---|---|---|---|---|
| | | | | upper | lower | min | Mean | max | WC94 | Le10 | |
| Psathopyrgos_fault | f1 | 8.5 | 60 | 0 | 6 | 4.6 | 5 | 5.5 | 5.8 | 5.7 | 2 kyr |
| Neos_Erineos_fault | f2 | 11.4 | 55 | 0 | 7 | 2.3 | 3.2 | 4.1 | 6.0 | 5.9 | 3 – 4 kyr |
| Aigion_fault | f3 | 8.6 | 60 | 0 | 7 | 3.5 | 4 | 4.6 | 5.8 | 5.8 | 50-60 kyr |
| East_Helike_fault | f4 | 14 .5 | 55 | 0 | 7 | 3 | 3.5 | 4 | 6.1 | 6.0 | 10-12 kyr |
| West_Helike_fault | f5 | 11.2 | 55 | 0 | 7 | 0.5 | 0.9 | 1.4 | 6.0 | 5.9 | 800 kyr |
| Trizonia_fault | f6 | 10.6 | 65 | 0 | 7 | 1.3 | 1.4 | 1.5 | 5.9 | 6.0 | 800 kyr |
| West_Channel_fault | f7 | 10.8 | 45 | 0 | 2.5 | 0.4 | 0.45 | 0.5 | 5.6 | 5.5 | 240 - 400 kyr |
| South_Eratini_fault | f8 | 12 | 45 | 0 | 6.5 | 0.6 | 1 | 1.4 | 6.0 | 6.0 | 800 kyr |
| East_Channel_fault | f9 | 22 | 45 | 0 | 4.5 | 1 | 1.4 | 1.8 | 5.7 | 5.7 | 1500 kyr |
| North_Erratini_fault | f10 | 11.5 | 60 | 0 | 6 | 2.4 | 4 | 5.6 | 5.9 | 5.8 | 12 kyr |
| Marathias_fault | f11 | 17.4 | 60 | 0 | 6.5 | 1.39 | 1.4 | 1.41 | 6.1 | 6.0 | 400 kyr |
| 1995_fault | f12 | 14 | 35 | 8 | 12 | 0.5 | 3.2 | 7 | 6.0 | 6.0 | 5 yr |
| Pyrgos_fault | f13 | 11 | 35 | 6 | 11 | 0.5 | 3.2 | 7 | 6.1 | 6.0 | 5 yr |

**Table 1: Fault characteristics in Boiselet, 2014. Mmax calculated using the equations for normal faults using the rupture area.**

| FtF sets | Faults involved in the scenario | | | | | Mmax | |
|---|---|---|---|---|---|---|---|
| | | | | | | WC94 | Le10 |
| B14, B14_hc, B14_s | All the single fault ruptures | | | | | see Table 1 | |
| **3 km distance criteria**<br><br>**B14**<br>**B14_hc** | f3 | f2 | | | | 6.2 | 6.2 |
| | f3 | f2 | f1 | | | 6.4 | 6.2 |
| | f2 | f1 | | | | 6.2 | 6.0 |
| | f4 | f5 | | | | 6.3 | 6.2 |
| | f1 | f13 | | | | 6.2 | 6.2 |
| | f4 | f12 | | | | 6.4 | 6.2 |
| | f4 | f8 | | | | 6.4 | 6.2 |
| | f4 | f8 | f5 | | | 6.5 | 6.4 |
| | f4 | f8 | f9 | | | 6.5 | 6.4 |
| | f8 | f9 | | | | 6.2 | 6.0 |
| **5 km distance criteria**<br><br><br><br>**B14_hc** | **f11** | **f6** | | | | **6.3** | **6.3** |
| | **f11** | **f6** | **f1** | | | **6.4** | **6.5** |
| | **f11** | **f6** | **f2** | | | **6.5** | **6.5** |
| | **f11** | **f6** | **f2** | **f1** | | **6.6** | **6.5** |
| | **f3** | **f5** | | | | **6.2** | **6.2** |
| | **f3** | **f7** | | | | **5.8** | **5.8** |
| | **f3** | **f9** | **f7** | | | **6.2** | **6.1** |
| | **f3** | **f8** | **f9** | **f7** | | **6.4** | **6.5** |
| | **f3** | **f4** | **f2** | **f1** | | **6.5** | **6.6** |
| | **f4** | **f7** | | | | **6.2** | **6.2** |
| | **f4** | **f8** | **f7** | | | **6.4** | **6.5** |
| | **f4** | **f8** | **f9** | **f7** | | **6.5** | **6.5** |
| | **f8** | **f10** | | | | **6.3** | **6.3** |
| | **f3** | **f6** | **f2** | **f1** | | **6.5** | **6.5** |
| | **f3** | **f12** | | | | **6.2** | **6.3** |
| | **f3** | **f4** | | | | **6.3** | **6.3** |
| | **f8** | **f9** | **f7** | | | **6.3** | **6.2** |
| | **f3** | **f4** | **f5** | **f2** | **f1** | **6.6** | **6.7** |

**Table 2: Ruptures scenarios considered in each model Branch B14_s considers only the single fault ruptures. Branch B14 considers the single fault ruptures and the FtF ruptures with a 3 km distance criteria. Branch B14-hc considers the single fault**

**ruptures and the FtF ruptures with a 3 km and the 5 km distance criteria. Mmax are calculated using the equations for normal faults based on the rupture area.**

| Date | Type of update | Old parameters | New parameters | Special implication for the catalogue |
|---|---|---|---|---|
| 1748 May 14 | Magnitude | M = 6.4 +- 0.25 | M = 5.9 [5.4 – 5.9] | Not in the complete period for this range of magnitudes |
| 1817 Aug 23 | Magnitude | M = 6.6 +- 0.25 | M = 6.5 [6.0 – 6.5] | |
| 1861 Dec 26 | Location | (38.22, 22.139) | (38.28, 22.24) | Not associated with Aigion fault |
| 1888 Sep 9 | Magnitude | M = 6.3 +- 0.4 | M = 6.2 [5.7 – 6.2] | |
| 1889 Aug 25 | Location and Magnitude | (38.25, 22.08) M = 6.24 +- 0.25 | (38.50, 21.33) M = 6.4 [6.4 – 6.6] | Earthquake outside the WCR |
| 1965 Mar 3 | Depth and Magnitude | Depth = 10 km M = 6.5 | Depth = 55 km M = 6.8 | Earthquake associated with the subduction zone, not with the WCR fault system |
| 1995 Jun 15 | Location and Magnitude | (38.37, 22.15) M = 5.8 | (38.36, 22.20) M = 6.3 | |
| 1997 Nov 05 | Location | (22.28,38.41) | (22.28,38.36) | |

5      **Table 3: Earthquakes updated in the historical and instrumental catalogues of the Western Corinth Rift**

| SHARE project | | Boiselet 2014 | |
|---|---|---|---|
| Magnitude range | Date of completeness | Magnitude range | Date of completeness |
| 4.1 – 5.1 | 1970 | 5.0 – 5.4 | 1958 |
| 5.1 – 5.7 | 1900 | 5.5 – 6.0 | 1904 |
| 5.7 – 6.5 | 1650 | 6.0 – 6.5 | 1725 |
| ≥ 6.5 | 1450 | 6.5 – 7.0 | 1725 |

**Table 4: Completeness hypothesis explored in this study.**

ANNEXE 1:

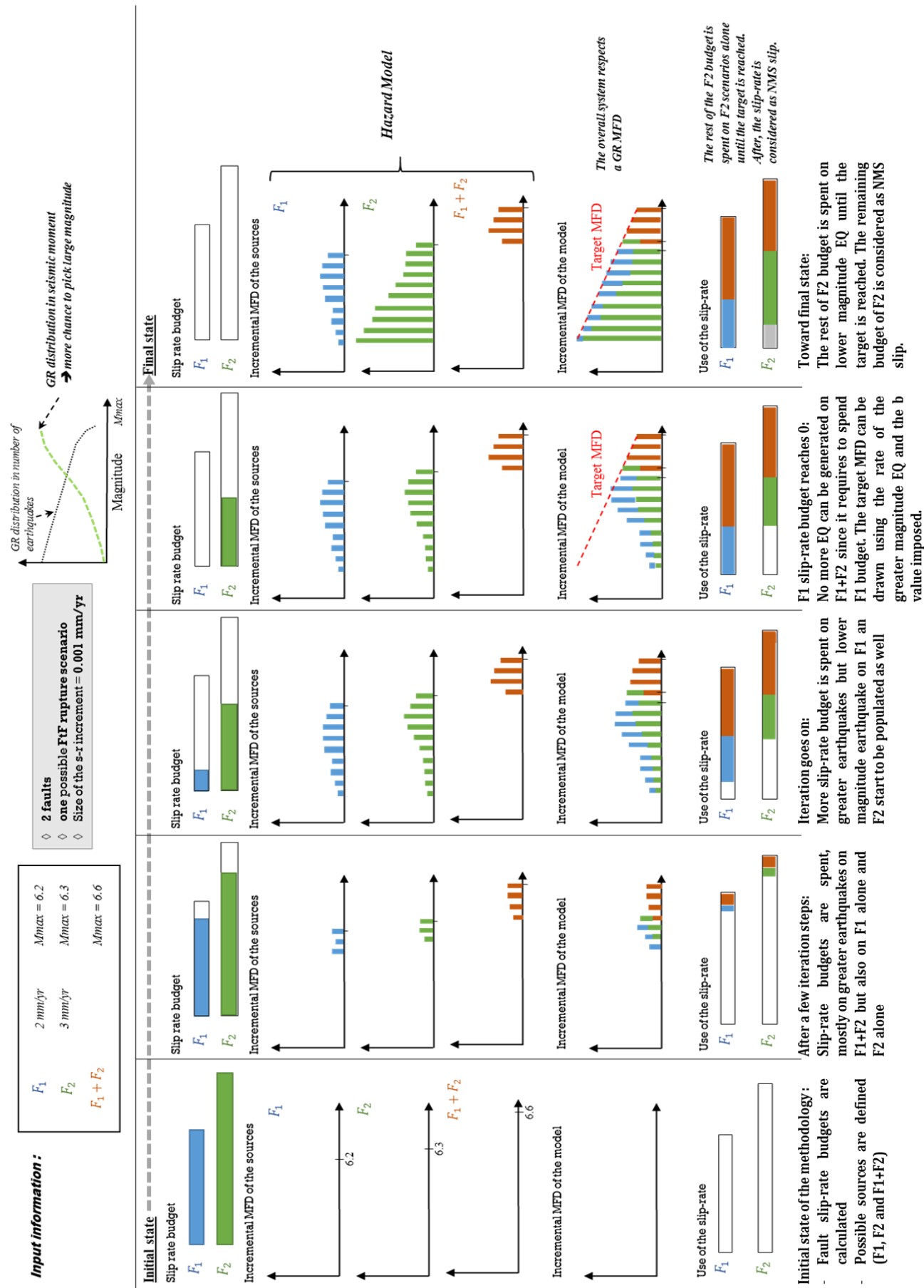