# Peer review of "Methodology for Earthquake Rupture Rate estimates of fault networks: example for the Western Corinth Rift, Greece"

_Natural Hazards and Earth System Sciences, 2017_

## Referee Comment (RC1) · K.-F. Ma (Referee) · 7 May 2017

It is indeed necessary and important task to propose a methodology to compute earthquake rate of the faults in the network for consideration in multiple fault segments rupture. This paper is scientific sounded and well presented. The methodology presented in this paper was tested on the Western Corinth Rift, Greece (WCR), which is in normal fault system regime. And, it is more distributed with moderate size earthquakes (M5.5∼6.5) historically. If it is possible, it would be good to discuss the pro and con of the methodology proposed in this paper to other faulting system, e.g. strike-slip or reverse. Or, this methodology might be limited only to the normal fault system, if so, why? As it is more in consideration of moderate size events, rather than other system,

which might giving larger Mmax as up to M>7 or larger?

Comments 1. The paper adopted the magnitude determination using Wells and Coppersmith (1994). This scaling is more scaled from strike-slip events from California. Is it also capable to the normal fault? Or, maybe to consider the fault area – magnitude relationship, which is more widely considered now in PSHA? Or, this cab be use din the logic tree. 2. The study in the Corinth Rift zone in a normal faulting system, historical events more in moderate earthquakes, what the implication this study can infer to other faulting system, as the strike-slip fault or collision fault system. 3. Terminology in using the words "subduction plane" in Table 2. What does that mean? Subduction zone interface events?

---

## Referee Comment (RC2) · Anonymous Referee #2 · 5 Jun 2017

**Review Comments for "Methodology for Earthquake Rupture Rate estimates of fault networks: example for the Western Corinth Rift, Greece" by Chartier et al. 2017.**

The manuscript proposes a new methodology to calculate the magnitude probability density function (PDF) for fault systems considering the fault-to-fault ruptures and seismic moment accumulation. Proposed methodology might be a good and repeatable alternative for the "grand inversion" technique used in UCERF3; therefore, I found the technical content of the manuscript important and worth to be published after the following issues are clarified:

1) The method proposed here is built on the assumption that the G-R distribution is valid for the faults and fault systems. Earthquakes in seismically active regions are observed to follow an exponential distribution of magnitudes (G-R distribution); however, the size distribution of earthquakes on faults has been the subject of debate. According to Hecker et al. (2013), the common practice in probabilistic seismic-hazard analysis (PSHA) is to favor a characteristic-earthquake distribution for faults, but to incorporate an exponential distribution in some aspect of the modeling. In Figure 1 of Hecker et al. (2013) a very clear example of the overestimation of the rates of small-to-moderate earthquakes when the G-R distribution applied to Hayward Fault is provided. Therefore, the authors should discuss the reasoning behind the selection of the target MFD as a G-R distribution. Would the Youngs and Coppersmith (1985) composite model be a better target MFD for fault-to-fault ruptures?

2) The computational steps for proposed methodology should be clearly demonstrated. Annex 1-Figure 1 is quite adequate for this purpose, but it is not properly explained in Section 2. Here is how I interpret the method from the text and Figure 1:
    a) Maximum magnitude and the magnitude bins for each fault are defined. For the example in Figure 1, $M_{max}$=6.2, 6.3, and 6.6 for F1, F2 and F1+F2, respectively.
    b) According to the figure, the computations start from the maximum magnitudes (Figure , panel 2). Since there is no other combination that can end-up in M=6.6, Mo for M=6.6 is calculated and reduced from F1+F2. Is my interpretation correct?
    c) The computations continue with decreasing magnitude. For the smaller M (for example M=6.2), all faults can be responsible. According to the text, the seismotectonic source that can be responsible for that is selected randomly. It can be F1, F2 or F1+F2. This point forward needs more explanation. What happens than? If F1 is randomly selected, the budgets for F1 and F1+F2 are both reduced? What happens to F2 e.g. can F2 also result in a magnitude 6.2 in this procedure?
    d) The incremental MFD on Figure 1 is equal to $dr_e/dM_o$. Is this correct? $dr_e/dM_o$ is basically equal to the seismic moment for that bin, coming from all fault combinations?
    e) Page 4, Line 6: "As the magnitude bins are picked according to a distribution based on the moment rate…" Can you please clarify that? Are the magnitude bins selected in a decreasing order (because the figure implies that)?
    f) Since the slip rates are spend in the decreasing order of magnitudes, this model somehow supports the characteristic assumption; the faults may not create small magnitude events if the budget is spent. This is consistent with Figure 3 third panel where the distribution looks like a skewed normal distribution. However, the rate of

the largest magnitude event ($dr_e/dM_o$) would be larger if $dr_e/dM_o=Mo(M=6.6)$. That's not consistent with Figure 1.

g) Page 4, Lines 9-11: "The target MFD for the whole fault-system is then calculated based on the imposed regional b value and the average rate of the three highest magnitude bins (0.3 being the range of uncertainties in the scaling laws used to assess the maximum magnitude)". To my understanding based on this statement and Figure 1, the activity rate (or the intercept of target MFD) is determined based on the known slope fitted to large magnitude rates. Can you please discuss the assumption that the slope is constant under the assumption that proposed model has a "close to characteristic" shape?

h) At the end, the shape achieved is "kind of" similar to the composite model of Young and Coppersmith (1985). Please discuss this similarity (or lack thereof) by plotting the proposed model and composite model in moment rate space.

Based on the questions raised above, the text explaining the procedure should be rewritten in more details for the sake of the reader, since it's the heart of the paper. Adding the spreadsheet for the example given in Figure 1 would also be very useful.

3) Proposed methodology does not have a check point. In the study referenced by the authors (Gülerce and Ocak, 2013), or in Hecker et al. (2013), assumed magnitude recurrence model is tested by the rate of earthquakes associated with that particular fault system for consistency. It seems like the authors foresee such a check point according to Figure 2 and 4. I recommend that the check is also added as the last step of the procedure.

4) Second part of the manuscript presents the application of the proposed methodology on western Corinth rift fault system. A few questions regarding the application side:

a) The b-value is assumed as 1.15. Please provide the reason why it is not calculated from the catalogue but assumed.

b) Page 5, Line 32: "We propagate the uncertainties on the earthquake magnitudes and on the time of completeness of the catalog in the seismic moment rate and earthquake rate calculations". Please explain this statement since the application procedure does not elaborate these matters.

c) I'm assuming that the catalogue completeness levels are considered in comparing the earthquake rates from the catalogue to the proposed MFD, specifically in Figure 4. Please clarify that issue.

d) One of the significant problems in utilizing the moment-balanced PSHA in the extensional regimes is the slip rate participation on parallel dipping faults (as in N. Erratini and S. Erratini Faults in Figure 2). Please explain how the extensional slip rate is calculated for these systems and how the uncertainty affects the proposed methodology.

e) Finally, the maximum magnitudes of the faults used in the example are not that big (none of them are above 6.5). Please comment on the applicability of this method for larger faults that can produce M>7 events.

---

## Author Comment (AC1) · 19 Jul 2017

Dear Professor Ma,

Thank you for your valuable questions and comments. Please consider our answers attached here and the modifications made to the article.

You will find attached a version of the article with "track change" and one without, probably easier to read.

On behalf of the authors,

Thomas Chartier

[Figure]

Please also note the supplement to this comment:
https://www.nat-hazards-earth-syst-sci-discuss.net/nhess-2017-124/nhess-2017-124-AC1-supplement.zip

―――――――――――――――

---

## Author Comment (AC2) · 19 Jul 2017

Dear Referee,

Thank you for your pertinent questions and comment. Please consider our answers attached here and the modifications made to the article.

You will find attached a version of the article with "track change" and one without, probably easier to read.

On behalf of the authors,

Thomas Chartier

[Figure]

Please also note the supplement to this comment:
https://www.nat-hazards-earth-syst-sci-discuss.net/nhess-2017-124/nhess-2017-124-AC2-supplement.zip

---

## Author Comment (AC3) · 24 Jul 2017

Dear Referee,

Please disregard the first answer that doesn't contain the correct attachment.

Thank you for your pertinent questions and comment. Please consider our answers attached here and the modifications made to the article. You will find attached a version of the article with "track change" and one without, probably easier to read.

On behalf of the authors,

Thomas Chartier

[Figure]

Please also note the supplement to this comment:
https://www.nat-hazards-earth-syst-sci-discuss.net/nhess-2017-124/nhess-2017-124-AC3-supplement.zip

————————————————————

---

## Author Response (AR1)

[revised manuscript text omitted]

ANNEXE 1:

---

## Editor Decision (ED1)

[revised manuscript text omitted]

*(margin annotation line 14–15)* Working Group of California Earthquake Probabilities, (WGCEP, 2003)

*(margin annotation line 29–30)* GR     geodetic?

2009, Napa earthquake 2014). These phenomena, called Non-Main-Shock slip (NMS) later on, are integrated in the slip-rates deduced from geological information and should not be converted into earthquake rate when computing seismic hazard. The methodology presented in this study allows part of the geological slip-rate to be considered as NSM slip-rate.

We use this methodology to generate fault-based hazard models for the Western Corinth Rift (WCR), Greece, which has been

5    studied for the past decade by the Corinth Rift Laboratory Working Group (CRL-WG) (Lyon-Caen et al, 2004; Bernard et al, 2006; Lambotte et al, 2014). A large number of active faults have been identified in this area and a consensus about their possible geometries and activity rates has been reached within the CRL-WG (Boiselet, 2014). We used this geologic information to test our modelling approach and explore different epistemic uncertainties in a logic tree. Finally, we confront the modelled earthquake rates of each fault with seismological and paleoseismological data in order to weigh the hypothesis

10   in the logic tree.

**2 Novel methodology for taking faults into account in PSHA**

In most regions of the world the amount of data available to model faults in a PSHA study is often sparse and uncertain. However, the need to consider such data in PSHA is increasing and the methods to properly incorporate the available geological information in the hazard models are still missing. In this  context, we propose to build a methodology that allows considering all

15   the available information on faults, allows setting rules to define FtF ruptures and considers single faults or FtF ruptures as an aleatory uncertainty. Our incremental method generates rates of earthquakes on faults by spending the slip-rate budget of each
 fault and following two rules: the resulting regional MFD of earthquakes in the whole  fault system follows an imposed shape and the rate of earthquakes on each fault is determined by the slip-rate on the fault. The shape of the individual MFD of each fault is not imposed.

20   The method requires a set of rupture scenarios as a list of the possible FtF ruptures in the fault model. In this study only a simple distance rule is used to define FtF ruptures. In future developments, more physics based approaches could be explored.

The proposed method is presented here in a nutshell and illustrated in Figure 1.

(1)  input data: are gathered, they include:

[Figure]

25       ▪    a definition of the 3D geometry of the fault system

         ▪    an estimate of the geological slip rates of each fault

(2)  hypothesis: Basic working ... are stated, namely:

         ▪    some suitable scaling laws to estimate the maximum magnitude each fault can host.

         ▪    the Minimum magnitude of earthquakes possible on the faults (5.0 in this study).

30       ▪    the possible FtF rupture scenarios.  in the example presented in Figure 1, three faults (*fault 1, fault 2* and *fault 3*) can rupture individually (F1, F2, F3) or in FtF rupture s (F1+F2, F2+F3 or F1+F2+F3)

         ▪    the shape of the  regional MFD  for the whole fault system. In this study a GR MFD distribution is assumed.

(3) Computational steps : are performed: they consist in:

[Figure]

35       ▪    Calculation of all possible magnitude bins each fault and FtF rupture scenario can accommodate according to the scaling law (Figure 1b).

         ▪     The slip-rate of each fault is spent in incremental quantities of slip-rate (*dsr*) that  along individual fault s or  FtF rupture scenario s. 
[revised manuscript text omitted]

ANNEXE 1: